# AV-PEA: Parameter-Efficient Adapter for Audio-Visual Multimodal Learning

## Abstract

Fine-tuning has emerged as a widely used transfer learning technique for leveraging pre-trained vision transformers in various downstream tasks. However, its success relies on tuning a significant number of trainable parameters, which could lead to significant costs in terms of both model training and storage. When it comes to audio-visual multimodal learning, the challenge also lies in effectively incorporating both audio and visual cues into the transfer learning process, especially when the original model has been trained with unimodal samples only. This paper introduces a novel audio-visual parameter-efficient adapter (AV-PEA) designed to improve multimodal transfer learning for audio-visual tasks. Through the integration of AV-PEA into a frozen vision transformer, like ViT (Dosovitskiy et al., 2021), the transformer becomes adept at processing audio inputs without prior knowledge of audio pre-training. This also facilitates the exchange of essential audio-visual cues between audio and visual modalities, all while introducing a limited set of trainable parameters into each block of the frozen transformer. The experimental results demonstrate that our AV-PEA consistently achieves superior or comparable performance to state-of-the-art methods in a range of audio-visual tasks, including audio-visual event localization (AVEL), audio-visual question answering (AVQA), audio-visual retrieval (AVR), and audio-visual captioning (AVC). Furthermore, it distinguishes itself from competitors by enabling seamless integration into these tasks while maintaining a consistent number of trainable parameters, typically accounting for less than 3.7% of the total parameters per task.

## 1 Introduction

The prevailing trend among recent machine learning models revolves around developing large-scale transformers to encode audio (Gong et al., 2021a; Chen et al., 2020), visual (Dosovitskiy et al., 2021; Radford et al., 2021), and language (Devlin et al., 2019) modalities. Recently, fine-tuning these large-scale pre-trained transformers (e.g. CLIP (Radford et al., 2021), BERT (Bugliarello et al., 2021), ViT (Dosovitskiy et al., 2021)) has also proven its high efficacy in achieving remarkable performance across various downstream tasks. The main advantage of transformers lies in their versatility, allowing seamless integration into various modalities with minimal modality-specific modifications required. This characteristic establishes a flexible and adjustable foundation for different data types, exemplified by transformers' current dominance as state-of-the-art (SOTA) models across several downstream tasks, to name a few, text-based image and video retrieval (Gabeur et al., 2022; Bugliarello et al., 2021; Ge et al., 2022), image and video captioning (Guo et al., 2022a; Yang et al., 2023a; Guo et al., 2022b), visual question answering (Shao et al., 2023; Ravi et al., 2023), and speech analysis (Grósz et al., 2022; Wang et al., 2023b).

Although large-scale models specialized for specific modalities like audio (Gong et al., 2021b), visual (Dosovitskiy et al., 2021), and text (Brown et al., 2020) often exhibit impressive performance on targeted tasks, they do encounter two significant limitations. First, optimizing and training models for a specific modality usually requires substantial computing resources (e.g. GPUs and memory) and relies heavily on extensive pre-trained datasets. For example, the GPT-3 (Brown et al., 2020) model requires 700GB of memory to accommodate its immense number of trainable parameters, which can reach up to 175 billion. This presents a challenge for smaller research laboratories with limited access to high-end computational capabilities (Sung et al., 2022). Second, fine-tuning such

large-scale models for downstream tasks using relatively small datasets can potentially lead to overfitting (Lin et al., 2023). The mismatch in scale between the model's capacity and the available downstream data may also impede the effective generalization of large-scale pre-trained models to new downstream tasks.

On the other hand, multimodal models aim to leverage correlations between different modalities, enabling a more comprehensive understanding of complex tasks that involve multiple sources of information, such as audio-visual event localization (AVEL) (Geng et al., 2023a; Xia & Zhao, 2022), audio-visual question answering (AVQA) (Li et al., 2022a; Yun et al., 2021), audio-visual retrieval (AVR) (Lin et al., 2022), and audio-visual captioning (AVC) (Chen et al., 2023). These models have gained significant attention due to their ability to handle real-world scenarios where data comes from diverse sources and often carries complementary information. An example of a large-scale audio-visual model is the multimodal bottleneck transformer (MBT) (Nagrani et al., 2021) which utilizes separate audio (Gong et al., 2021b) and visual (Dosovitskiy et al., 2021) transformers, trained independently on their respective modalities, before integrates them through late fusion techniques harnessing the benefits of cross-modality interactions. However, late fusion techniques often fail to leverage cross-modal cues in the early layers, leading to suboptimal performance in audio-visual tasks requiring integrated multimodal reasoning. Additionally, this necessitates separate audio and visual dataset curation during pre-training, imposing significant memory and GPU resource demands.

On top of all these, transformers are continuously growing in size, making full fine-tuning increasingly infeasible. To address these challenges, parameter-efficient fine-tuning approaches, such as prompt tuning (Kirillov et al., 2023; Wasim et al., 2023) and adapter modules (Houlsby et al., 2019; Karimi Mahabadi et al., 2021; Lin et al., 2023; Sung et al., 2022; Pan et al., 2022), have emerged as a solution. Among these approaches, adapter modules have demonstrated excellent performance by introducing a limited set of trainable parameters while keeping the pre-trained model parameters frozen (Houlsby et al., 2019; Lin et al., 2023; Sung et al., 2022; Pan et al., 2022). Freezing the pre-trained model's parameters allows effective transfer of knowledge gained from a large-scale pre-training dataset to downstream tasks. Moreover, these frozen parameters can be readily shared among different modalities (e.g. audio and visual). This approach not only optimizes resource utilization, but also encourages seamless transfer of knowledge between these distinct modalities (Houlsby et al., 2019; Lin et al., 2023). Drawing inspiration from the adaptability of the transformer architecture, which can be applied to diverse modalities with minimal modality-specific alterations, we find examples such as the BERT language transformer (Devlin et al., 2019) being extensively used in a wide range of domains. These domains span image and video processing (Li et al., 2022b; Wang et al., 2022), and speech analysis (Hsu et al., 2021; Chang et al., 2022).

The main goal of this work is to investigate the capacity of pre-trained vision transformers to generalize across diverse multimodal domains, with a specific emphasis on the field of audio-visual learning. In this context, the core idea revolves around the representation of audio inputs as 2D spectrogram images, which can be jointly processed alongside real visual inputs using a vision transformer. This approach eliminates the need for prior pre-training of the transformer on a separate audio dataset. To achieve this goal, we propose an innovative audio-visual parameter-efficient adapter (AV-PEA) explicitly crafted for multimodal learning. The proposed AV-PEA facilitates seamless adaptation of frozen vision transformers, initially pre-trained on images, to audio-visual tasks. It also effectively leverages the complementary nature of audio and visual modalities through a cross-attention module, all achieved with a limited set of extra trainable parameters. Specifically, within a dual-stream visual transformer, AV-PEA is employed at each layer to enhance the representations of both audio and visual inputs. This enhancement is achieved through a proficient cross-attention module, followed by a lightweight bottleneck block, wherein each stream generates a token dedicated to facilitating information exchange with the other stream. By utilizing a single token from each stream for information exchange, it significantly mitigates the quadratic costs typically associated with traditional cross-attention mechanisms, resulting in enhanced overall efficiency.

The key contributions of our work are outlined as follows: (a) We propose a novel adapter, called AV-PEA, to adapt pre-trained vision transformers for efficient audio learning without requiring an audio model pre-trained with a large dataset. (b) We introduce a simple yet effective token fusion module founded on cross-attention, which operates linearly in both computation and memory usage while effectively improving the integration of cues from both audio and visual modalities. (c) Our AV-PEA outperforms contemporary audio-visual adapter modules in terms of accuracy and model

parameters, while also achieving performance on par with or exceeding SOTA methods in various audio-visual downstream tasks, such as AVEL, AVQA, AVR, and AVC.

## 2  RELATED WORK

**Audio-Visual Pre-trained Models**. Vision transformer (ViT) (Dosovitskiy et al., 2021) and audio spectrogram transformer (AST) (Gong et al., 2021b) have emerged as cutting-edge solutions for image and audio classification, respectively. Beyond their original specific tasks, these models have shown significant potential as versatile foundations for transfer learning in various downstream tasks (Chen et al., 2023). Typically, they undergo training using extensive labeled datasets (such as ImageNet (Deng et al., 2009) and AudioSet (Gemmeke et al., 2017)) in a supervised manner. However, recent models (Radford et al., 2021; Wang et al., 2023a; Guzhov et al., 2022) have embraced multimodal data (e.g. audio-visual and text pairs, image-text pairs, and video-text pairs) resulting in more potent representations.

**Audio-Visual Learning**. Audio-visual learning tasks evolve on the integration and understanding of information from both audio and visual modalities. These tasks often involve processing data that includes both audio signals, such as speech or sound (Gong et al., 2022; Lin et al., 2022), and visual cues, such as images or videos. The goal is to leverage the complementary information from both modalities to achieve improved performance in various tasks, including but not limited to AVEL (Tian et al., 2018; Xia & Zhao, 2022), AVQA (Li et al., 2022a; Yun et al., 2021), AVR (Chen et al., 2023; Li et al., 2022a; Yun et al., 2021), AVC (Chen et al., 2023). The AVEL task involves identifying and localizing events within a multimedia context (e.g. video) that are observable in both audio and visual data. This involves not only identifying when an event occurs, but also precisely delineating its temporal boundaries (Tian et al., 2018; Geng et al., 2023b). The majority of current methods (Tian et al., 2018; Rao et al., 2022; Xia & Zhao, 2022) developed for AVEL tasks in the literature depend on pre-trained audio and visual models (e.g. VGGish (Hershey et al., 2017) and ResNet-152 (He et al., 2016)) tailored to each modality. These models are employed to extract distinct audio and visual features, which are subsequently integrated to facilitate AVEL. AVQA is a task that combines both audio and visual modalities with natural language processing to answer human-generated questions concerning audio-visual content. Similar to the context of AVEL tasks, a significant portion of existing methods designed for the AVQA task relies on audio and vision models specialized for their respective modalities. These models are then merged through spatial and temporal grounding modules (Yun et al., 2021) to effectively provide meaningful answer. However, in such contexts, irrelevant audio and visual elements processed by modality-specific models may introduce learning noise, adding complexity to the task. The AVR task involves retrieving relevant multimedia content (i.e. images, videos, or audio clips) based on a query that consists of both audio and visual input, while the AVC task involves crafting informative textual captions for multimedia content that includes both audio and visual elements. Recently, Chen et al. (2023) introduced VALOR, a novel tri-modality (Vision-Audio-Language) pre-trained model and dataset designed to evaluate audiovisual-language capabilities, including tasks like AVR and AVC. Notably, the VALOR pre-trained model is also built upon the ViT framework.

**Parameter-Efficient Transfer Learning (PETL)**. The PETL principle has been introduced in the domain of natural language processing to mitigate the escalating computational demands associated with full fine-tuning of ever-growing language models across diverse downstream tasks. This is achieved either by introducing a set of trainable tokens (prompt tuning) at the input (Wasim et al., 2023) or by incorporating lightweight modules (adapters) between the layers of a pre-trained model (Houlsby et al., 2019; Pfeiffer et al., 2020). In the same context, PETL has gained significant traction in the computer vision (CV) domain, as evidenced by recent works (Karimi Mahabadi et al., 2021; Sung et al., 2022; Pan et al., 2022; Yang et al., 2023b; Lin et al., 2023; Ju et al., 2022; Kirillov et al., 2023). Sung et al. (2022) developed a vision-language adapter module that targets the text encoder of the CLIP model. Recently, Pan et al. (2022) and Yang et al. (2023b) proposed adapter modules to adapt pre-trained image transformer models for video understanding, concentrating on the video action recognition research. Concurrently, there has been a growing interest in the exploration of prompt tuning techniques to enhance visual transformers, as demonstrated by the works of Kirillov et al. (2023) and Ju et al. (2022).

However, most existing adapter modules in the literature are designed for specific tasks and often lack the ability to effectively facilitate cross-modal information exchange. To the best of our knowledge, the latent audio-visual hybrid (LAVISH) adapter (Lin et al., 2023) stands as a singular instance of PETL modules developed for audio-visual learning. The LAVISH adapter utilizes a compact collection of latent tokens to first compress information from all modality-specific tokens (i.e. audio and video). It subsequently applies cross-attention between these latent tokens and all tokens from another modality. This enables a two-way flow of information between the audio and video modalities, leading to an enhanced audio-visual representation. Nonetheless, significant distinctions exist between LAVISH and our AV-PEA. First, LAVISH requires the adjustment of its hyper-parameters for each new audio-visual downstream task. In contrast, our AV-PEA seamlessly integrates into novel audio-visual tasks with a consistent design and invariant parameters, while enjoying better performance and less trainable parameters. Second, LAVISH relies on latent tokens, which are heavily influenced by the downstream dataset size, for facilitating information exchange between audio and visual modalities. Conversely, our AV-PEA relies exclusively on the $CLS$ token from each modality for cross-modal information exchange, regardless of the downstream dataset size.

## 3 METHOD

In this section, we propose AV-PEA, a novel audio-visual adapter designed to fine-tune frozen pre-trained large-scale vision transformers (e.g. ViT (Dosovitskiy et al., 2021)) for various audio-visual downstream tasks (like AVEL, AVQA, AVR, and AVC), while introducing only a limited set of trainable parameters. We will begin with a concise overview of ViT as an example of a transformer capable of accommodating the proposed AV-PEA adapter, and then present the AV-PEA approach. Finally, we will delve into the technical details of seamlessly integrating AV-PEA into the ViT transformer.

### 3.1 VIT TRANSFORMER

ViT draws inspiration from natural language processing transformers, like BERT (Devlin et al., 2019), to capture complex relationships among visual components through self-attention mechanisms. This model has gained significant prominence in the field of computer vision, attracting considerable interest and consistently delivering exceptional classification performance. In ViT (Figure 1a), the initial step involves transforming the input image into fixed-size patches, known as tokens, through the ViT's embedding layer. Similar to the BERT model, an additional classification ($CLS$) token is introduced among the image patch tokens to represent the global context of the image. To capture spatial relationships, position embeddings are also integrated into each token, providing crucial positional information. These tokens are then directed into a series of stacked transformer blocks for further processing. Each transformer block consists of a multiheaded self-attention (MSA) layer and a feed-forward network (FFN), collectively enhancing the model's ability to capture and integrate pertinent visual information across the entire sequence of token. Finally, the classification task is performed using the information aggregated within the $CLS$ token (Dosovitskiy et al., 2021; Chen et al., 2021).

### 3.2 THE PROPOSED AV-PEA

Our AV-PEA is founded on a parameter-efficient bottleneck block, as introduced by Houlsby et al. (2019). This bottleneck block is applied on top of a simple cross-attention (CA) module as shown in Figure 1b. Particularly, our AV-PEA capitalizes on the ability of the $CLS$ token in ViT to capture abstract information among patch tokens, thus enhancing audio-visual representation through the CA module. To achieve this, we propose a dual-stream ViT transformer (Figure 1a): the *visual-stream* for processing visual input and the *audio-stream* for processing audio input. Within each block of both streams, we integrate our AV-PEA to efficiently adapt the ViT transformer to audio input (which is unseen during the training phase of ViT) while also enabling seamless information exchange between the audio and visual streams. In the CA module, the $CLS$ token of each stream serves as an intermediary to facilitate information exchange with the token sequence from the other stream. The $CLS$ token is then back-projected to its respective stream, allowing it to interact with its own patch tokens once again in the bottleneck block. This enables the learned information from

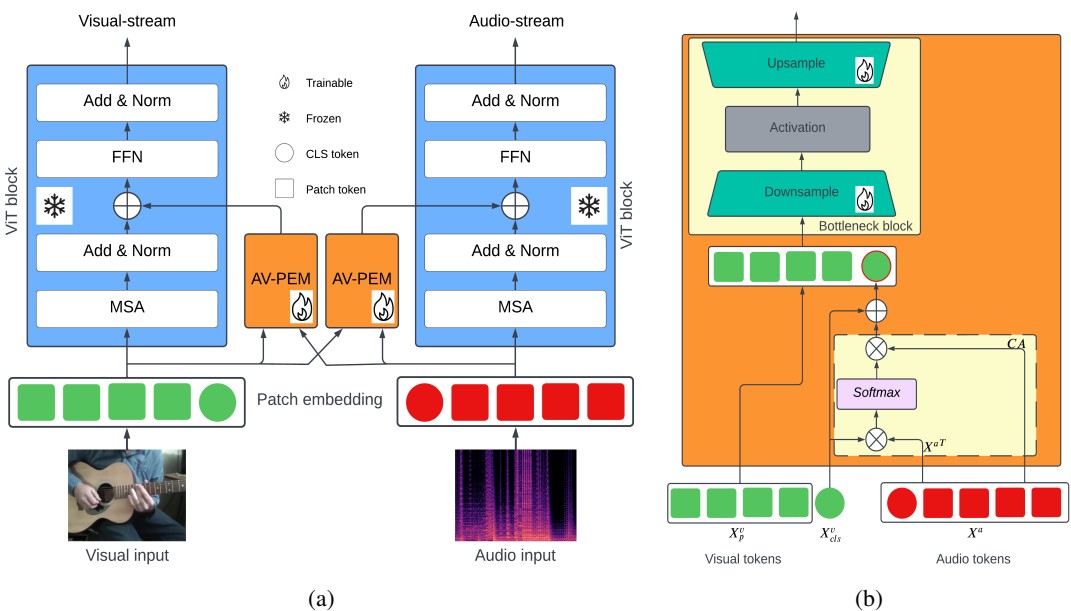

Figure 1: (a) Integration of the proposed AV-PEA into the ViT transformer. (b) The proposed AV-PEA, highlighting the cross-attention (CA) module enclosed by a dotted rectangle.

the other stream to be effectively conveyed to each patch token, thereby enriching the representation of individual patch tokens and ensuring comprehensive integration of multimodal representations.

### 3.3 TECHNICAL INTEGRATION OF AV-PEA INTO THE VIT TRANSFORMER

Within our proposed dual-stream ViT transformer (Figure 1a), consider the visual tokens $X^v \in \mathbb{R}^{(n+1) \times D}$, comprising both the patch tokens $X_p^v \in \mathbb{R}^{n \times D}$ and the $CLS$ token $X_{cls}^v \in \mathbb{R}^{1 \times D}$ directed to the visual stream. Similarly, the audio tokens $X^a \in \mathbb{R}^{(n+1) \times D}$ consist of the patch tokens $X_p^a \in \mathbb{R}^{n \times D}$ and the $CLS$ token $X_{cls}^a \in \mathbb{R}^{1 \times D}$ directed to the audio stream, where $n$ and $D$ represent the number of patch tokens and the embedding dimension, respectively. Before we integrate our AV-PEA into the ViT block of each stream, let's first outline the standard operations of a ViT block $\ell$ within the visual stream $v$. The block $\ell$ begins by applying the multiheaded self-attention layer (MSA) as:

$$Y_\ell^v = X_\ell^v + \text{MSA}(X_\ell^v). \tag{1}$$

Subsequently, the intermediate representation $Y_\ell^v$ from MSA is passed through the feed-forward network (FFN) of the block $\ell$, resulting in:

$$X_{\ell+1}^v = Y_\ell^v + \text{FFN}(Y_\ell^v). \tag{2}$$

These MSA and FFN operations are iteratively applied to the visual tokens $X^v$ at each block of $v$. The same procedure is applied to the audio stream $a$, with the only difference being the interchange of the indices $v$ and $a$.

The integration of AV-PEA into each block $\ell$ of the dual-stream ViT transformer proceeds as follows:

$$X_{\ell+1}^v = Y_\ell^v + \text{FFN}(Y_\ell^v) \qquad \text{and} \qquad Y_\ell^v = X_\ell^v + \text{MSA}(X_\ell^v) + B_\ell^v \tag{3}$$

$$X_{\ell+1}^a = Y_\ell^a + \text{FFN}(Y_\ell^a) \qquad \text{and} \qquad Y_\ell^a = X_\ell^a + \text{MSA}(X_\ell^a) + B_\ell^a, \tag{4}$$

where $B_\ell^v$ and $B_\ell^a$ denote the bottleneck blocks of AV-PEA on the $v$ and $a$ streams, respectively. Mathematically, the expressions for the $B_\ell^v$ and $B_\ell^a$ bottleneck blocks are as follows:

$$B_\ell^v = h_v \cdot f^v(CA_v \parallel X_p^v) \tag{5}$$

$$B_\ell^a = h_a \cdot f^a(CA_a \parallel X_p^a), \tag{6}$$

where $f$ is the projection function of the bottleneck block, $\parallel$ denotes concatenation, and $h$ is a scalar trainable parameter that acts as a learnable gate to regulate the flow of information through the model. The $CA_v$ and $CA_a$ denote the cross-attention process within the AV-PEA of the $v$ and $a$ streams, respectively, and can be mathematically expressed as follows:

$$CA_v(X_{cls}^v, X^a) = g_v \cdot \Theta_v X^a \qquad \text{and} \qquad \Theta_v = Softmax(X_{cls}^v X^{aT}) \tag{7}$$

$$CA_a(X_{cls}^a, X^v) = g_a \cdot \Theta_a X^v \qquad \text{and} \qquad \Theta_a = Softmax(X_{cls}^a X^{vT}), \tag{8}$$

where $g$ is a scalar trainable parameter utilized to control the flow of information between the two streams. Equations 7 and 8 reveal that only the $CLS$ token is used as the query, ensuring that the generation of the attention maps $\Theta$ maintain linear computation and memory complexity. In addition to the CA process, the bottleneck block in AV-PEA involves projecting the original $D$-dimensional tokens into a lower-dimensional space with dimensionality $d$. Subsequently, a non-linear activation function *ReLU* is applied before projecting the tokens back into their original $D$-dimensional space. This dimensionality reduction, achieved by setting $d \ll D$, substantially decreases the number of additional parameters.

## 4 EXPERIMENTS

### 4.1 EXPERIMENTAL SETTINGS

For the AVEL and AVQA experiments: we employed the conventional ViT (Dosovitskiy et al., 2021) model, which underwent supervised pre-training on annotated data sourced from ImageNet-21K (Deng et al., 2009) as our base pre-trained model. The ViT-B/16 and ViT-L/16 variants, optimized for processing patches of size $16 \times 16$, took precedence in most of our experiments. In the context of the AVR and AVC experiments, we integrated our AV-PEA into the VALOR pre-trained model (Chen et al., 2023). While this model shares foundational principles with the ViT transformer, it has undergone supervised pre-training on the VALOR-1M audio-visual-language dataset (Chen et al., 2023). To conduct a comprehensive comparison with the SOTA models, we just replaced the visual and audio encoders of the SOTA models with the frozen ViT (or VALOR) transformer augmented by our AV-PEA, as explained in Section3. Additionally, we followed the evaluation procedures of the SOTA approaches, including the extraction of audio and visual features, to ensure methodological alignment. Throughout the training process, the parameters of the pre-trained transformer remained frozen, while the parameters of the AV-PEA were randomly initialized to meet the specific requirements of the audio-visual downstream task. Across all our experiments, we maintained a consistent learning rate of $3 \times 10^{-4}$, set $D$ to eight times $d$, and initialized $g$, $h_a$, and $h_v$ from zero.

### 4.2 DOWNSTREAM TASKS AND RESULTS ANALYSIS

**AVEL:** the audio-visual event (AVE) dataset (Tian et al., 2018) was used to assess the performance of our AV-PEA within the audio-visual event localization task. This dataset consists of 4,143 fully-supervised videos, with 3,339 in the training set, 402 in the validation set, and 402 in the testing set. Each video lasts 10 seconds covering events belonging to 28 distinct categories. To this end, AV-PEA was incorporated into the cross-modal background suppression (CMBS) model (Xia & Zhao, 2022) with replacing its pre-trained visual and audio encoders by the frozen ViT transformer. Following the procedure outlined in the CMBS work (Xia & Zhao, 2022), the event category label for each second within the videos was predicted, and the model's performance was evaluated using the overall accuracy metric for predicting event categories. The comparison results with SOTA models on the AVE dataset were presented in Table 1. Our primary emphasis was placed on the CMBS model,

Table 1: Audio-Visual Event Localization (AVEL): comparison with SOTA on the AVE dateset. Within this context, "PD" stands for pre-trained dataset, "N/A" abbreviates not available, ⋆ indicates the absence of official code, ✗ denotes a non-relevance criterion, ❄ signifies frozen, and 🔥 means full fine-tuning.

| | | | | | Parameters (M)↓ | | | |
| | | | | | Adapter | Total | | |
| Method | Visual Encoder | Audio Encoder | Visual PD | Audio PD | 🔥 | 🔥 | ❄ | Acc↑ |
|---|---|---|---|---|---|---|---|---|
| DPNet⋆Rao et al. (2022) | VGG-19 | VGGish | ImageNet | AudioSet | ✗ | N/A | N/A | 79.68 |
| CMBSXia & Zhao (2022) | ResNet-152❄ | VGGish❄ | ImageNet | AudioSet | ✗ | 14.4 | 202.3 | 79.70 |
| MBTNagrani et al. (2021) | ViT-B/16🔥 | AST🔥 | ImageNet | AudioSet | ✗ | 172 | ✗ | 77.80 |
| LAVISHLin et al. (2023) | ViT-B/16❄ (shared) | | ImageNet | ✗ | 3.9 | 4.7 | 102.5 | 75.30 |
| LAVISH Lin et al. (2023) | ViT-L/16❄ (shared) | | ImageNet | ✗ | 13.4 | 14.5 | 325.6 | 78.10 |
| CMBS+AV-PEA(Ours) | ViT-B/16❄ (shared) | | ImageNet | ✗ | 3.7 | 17.8 | 102.5 | 75.65 |
| CMBS+AV-PEA(Ours) | ViT-L/16❄ (shared) | | ImageNet | ✗ | 12.9 | 27.2 | 325.6 | 79.90 |

well-known for its attainment of SOTA results on the AVE benchmark dataset. Furthermore, we conducted comparative analyses with the published outcomes derived from the MBT, the recent LAVISH adapter, and the dual perspective network (DPNet) (Rao et al., 2022) on the AVE dataset. Importantly, the LAVISH adapter employed the same pre-trained ViT models as those integrated with our AV-PEA.

From Table 1, among the models employing AudioSet pre-training and demanding modality-specific dual encoders (visual and audio), the MBT model demonstrated the lowest accuracy (77.80%), lagging behind both DPNet and CMBS (79.68% and 79.70%, respectively). This is a significant observation, especially considering that the MBT model underwent full parameter tuning. Without the need for extensive audio pre-training on AudioSet, the LAVISH and our AV-PEA approaches, based on ViT-B and utilizing a shared pre-trained encoder for both visual and audio inputs, achieved comparable results ranging from 75.30% to 75.65%. However, our AV-PEA achieved this while utilizing fewer adapter parameters than LAVISH (3.7M vs. 3.9M), and amounting to just 3.1% of the total parameters (3.7M vs. (17.8+102.5)M). Significantly, our AV-PEA with ViT-L outperformed all other methods, attaining an accuracy of 79.90%, even surpassing the analogous LAVISH adapter with ViT-L (78.10%). Worth noting is that LAVISH presented lower performance on larger models like ViT-L due to its substantial reliance on latent tokens. On the contrary, our AV-PEA model demonstrated continuous improvement, all while utilizing fewer adapter parameters than LAVISH (12.9M vs. 13.4M), accounting for only 3.7% of the total parameters (12.9M vs. (27.2+325.6)M), all the while capitalizing on its seamless plug-and-play functionality.

**AVQA:** In Table 2, we further evaluated the effectiveness of our AV-PEA in the context of audio-visual question answering task, utilizing the MUSIC-AVQA (Li et al., 2022a) dataset. In these experiments, we implemented a more robust AVQA (Li et al., 2022a) baseline using the frozen ViT augmented with our AV-PEA. The MUSIC-AVQA dataset comprises 9,288 videos and 45,867 question-answer pairs. It includes 33 question templates encompassing 9 question types, which span across audio, visual, and audio-visual domains. Each of these question templates is associated with a specific answer, resulting in a pool of 42 potential answers. The dataset is divided into training, validation, and testing sets containing 32,087, 4,595, and 9,185 QA pairs, respectively.

Table 2 revealed outstanding performance by the AVQA (Li et al., 2022a) with Swin-V2-L visual encoder within the methods employing AudioSet pre-training. This configuration of AVQA achieved a marginal accuracy improvement of 0.84% compared to the baseline AVQA (Li et al., 2022a) employing a ResNet-1 visual encoder. However, achieving this modest improvement demanded the integration of an extra 229.4M trainable parameters. These experiments also highlight the limitations of the LAVISH adapter with larger datasets such as the MUSIC-AVQA dataset. Remarkably, LAVISH with ViT-B/16 presented inferior performance compared to its own baseline AVQA model (68.93% vs. 73.37%). This is despite the introduction of additional latent tokens, as evidenced by the contrast in the number of adapter parameters of the AVEL (Table 1) and AVQA (Table 2) tasks (3.9M vs. 4.4M). On the contrary, our AV-PEA with ViT-B/16 not only outperformed AVSD (Schwartz et al., 2019) and Pano-AVQA (Yun et al., 2021), but also surpassed various AVQA baseline variants, including LAVISH with ViT-B/16. Additionally, it obtained comparable results to the LAVISH with ViT-L/16 (74.90% vs. 74.94%), while utilizing only 0.25 of trainable parameters used by the LAVISH with ViT-L/16. Finally, we noted a consistent improvement in accuracy through our

Table 2: Audio-Visual Question Answering (AVQA) using the Music-AVQA dataset. We reported accuracy spans three question categories: audio, visual, and audio-visual.

| Method | Visual Encoder | Audio Encoder | Visual PD | Audio PD | Parameters (M) ↓ | | | Question ↑ | | | |
| | | | | | Adapter 🔥 | Total 🔥 | Total ❄ | Audio | Visual | Audio-visual | Avg ↑ |
|---|---|---|---|---|---|---|---|---|---|---|---|
| AVSD*Schwartz et al. (2019) | VGG-19 | VGGish | ImageNet | AudioSet | ✗ | N/A | N/A | 68.52 | 70.83 | 65.49 | 68.28 |
| Pano-AVQA*Yun et al. (2021) | Faster RCNN | VGGish | ImageNet | AudioSet | ✗ | N/A | N/A | 70.73 | 72.56 | 66.64 | 69.98 |
| AVQALi et al. (2022a) | ResNet-18 ❄ | VGGish ❄ | ImageNet | AudioSet | ✗ | 10.6 | 94.4 | 74.06 | 74.00 | 69.54 | 72.53 |
| AVQALi et al. (2022a) | Swin-V2-L 🔥 | VGGish ❄ | ImageNet | AudioSet | ✗ | 240 | 312.1 | 73.16 | 73.80 | 73.16 | 73.37 |
| AVQA+LAVISH | ViT-B/16 ❄ (shared) | | ImageNet | ✗ | 4.4 | 13.1 | 102.5 | 73.14 | 68.73 | 64.93 | 68.93 |
| AVQA+LAVISH | ViT-L/16 ❄ (shared) | | ImageNet | ✗ | 14.8 | 23.8 | 325.6 | 75.05 | 79.44 | 70.34 | 74.94 |
| AVQA+AV-PEA(Ours) | ViT-B/16 ❄ (shared) | | ImageNet | ✗ | 3.7 | 12.4 | 102.5 | 76.16 | 78.82 | 69.72 | 74.90 |
| AVQA+AV-PEA(Ours) | ViT-L/16 ❄ (shared) | | ImageNet | ✗ | 12.9 | 21.9 | 325.6 | 74.49 | 80.06 | 71.26 | 75.27 |

AV-PEA with ViT-L/16, achieving an accuracy of 75.27%, and amounting to just 3.7% of the total parameters (12.9M vs. (21.9+325.6)M). It's noteworthy that our AV-PEA adapter maintains parameter consistency across diverse tasks, coupled with its user-friendly design that enables effortless integration into new tasks, eliminating the need for parameter adjustments.

**AVR and AVC:** thanks to the seamless design of our AV-PEA, it allows for easy integration into pre-trained models across various downstream tasks. For the audio-visual retrieval and captioning tasks, our AV-PEA was incorporated into the recent VALOR pre-trained model, and subsequently evaluated using the VALOR-32K (Chen et al., 2023) dataset. The VALOR-32K dataset includes 32K videos (25K for training, 3.5K for validation, and 3.5K for testing), and serves as a vision-audio-language correlated dataset specifically designed for tri-modality downstream tasks. For a fair comparison with the rival LAVISH, we integrated the LAVISH adapter into the frozen VALOR model. Specifically, we replaced the audio transformer of VALOR with its corresponding frozen visual transformer, thereby excluding the need for AudioSet pre-training. Just like the VALOR evaluation protocol, the recall at rank $K$ ($R@K, K = 1, 5, 10$) were used as metrics for the AVR task, whereas BLEU4, METEOR, and ROUGE-L were used as metrics for the AVC task. On top of these, our evaluation extended to re-evaluating the performance of both the AV-PEA and LAVISH approach, now integrated into the VALOR model, using the MUSIC-AVQA dataset. This evaluation was conducted in line with the VALOR framework. Worth noting is that while the AVQA framework in Table 2 primarily pertains to a classification problem where answers are retrieved from a pool of 42 potential answers, the VALOR framework formulates the AVQA task as a generative problem, aiming to directly generate the answer based on the input question.

The results presented in Table 3 revealed several findings. Firstly, our AV-PEA presented superior average performance in comparison to the baseline VALOR model for the AVC task (22.51 vs. 18.93), despite not using a pre-trained audio encoder or undergoing extensive AudioSet pre-training like the VALOR model. Secondly, our AV-PEA performed comparably to the VALOR model for the AVQA task (78.63% and 78.90%). Thirdly, our AV-PEA showcased a slight performance improvement over the LAVISH for both the AVC (22.51 vs. 22.41) and AVQA (78.63% vs. 77.93%) tasks, while maintained parity on the AVR task (81.00% and 81.10%). Finally, it's truly impressive to witness the remarkable efficacy of adapter modules, including our AV-PEA and the LAVISH, when seamlessly incorporated into pre-trained models. Even with a relatively modest count of additional trainable parameters and without the need for extensive AudioSet pre-training, these adapter modules manage to attain comparable or even superior performance across a range of downstream tasks.

Table 3: Comparison of performance results on the VALOR-32K dataset, covering Text-to-Audio-Visual Retrieval (AVR) and Audio-Visual Captioning (AVC), along with results on the MUSIC-AVQA dataset, which focuses on the Audio-Visual Question Answering (AVQA) benchmark.

| Method | AVR ↑ | | | | AVC ↑ | | | | AVQA ↑ |
| | R@1 | R@5 | R@10 | Avg | BLEU4 | METEOR | ROUGE-L | Avg | Acc |
|---|---|---|---|---|---|---|---|---|---|
| VALOR | 67.90 | 89.70 | 94.40 | 84.00 | 9.60 | 15.40 | 31.80 | 18.93 | 78.90 |
| VALOR+LAVISH | 64.70 | 86.70 | 92.00 | 81.10 | 11.14 | 19.53 | 36.66 | 22.44 | 77.93 |
| VALOR+AV-PEA(Ours) | 64.10 | 86.60 | 92.40 | 81.00 | 11.37 | 19.09 | 37.06 | 22.51 | 78.63 |

Table 4: Effectiveness of AV-PEA on audio-visual learning.

| Method | Audio stream | Visual stream | Acc ↑ |
|--------|:------------:|:-------------:|:-----:|
| CMBS | ✗ | ✗ | 72.01 |
| CMBS | AV-PEA | ✗ | 72.71 |
| CMBS | ✗ | AV-PEA | 74.68 |
| CMBS | AV-PEA | AV-PEA | 75.65 |

### 4.3 ABLATION STUDIES

To validate the efficiency of our AV-PEA in the context of the dual-stream transformer (Figure 1), we used the ViT-B/16 pre-trained model on the AVE dataset (Tian et al., 2018). We replaced the visual and audio encoders of the CMBS (Xia & Zhao, 2022) model with the frozen ViT-B/16 transformer, and integrated our AV-PEA into each transformer block following the methodology detailed in Section 3.3. We delved into a range of different design possibilities for our AV-PEA. This encompassed scenarios where the AV-PEA was integrated into both the visual and audio streams, as well as instances where it was omitted from either of them.

As observed in Table 4, AV-PEA played a significant role in bridging the gap in handling audio inputs, as evident from the results achieved through the integration of AV-PEA on the audio stream (72.71% vs. 72.01%). This was achieved despite the frozen ViT pre-trained model did not undergo AudioSet pre-training. It also demonstrated significant enhancement in the visual stream (74.68% vs. 72.01%), primarily attributed to the CA module (Figure 1b), which effectively enables the exchange of information between the audio and visual modalities, leading to the robust establishment of audio-visual cues in both streams. Last but not least, it becomes evident that integrating AV-PEA into both the audio and visual streams clearly outperforms the highest achievement obtained by augmenting only the visual stream with AV-PEA (75.65% vs. 74.68%).

## 5 CONCLUSIONS

In this paper, we introduced a novel audio-visual parameter-efficient adapter (AV-PEA) module that serves a dual purpose: (1) simplifying the integration of audio inputs into frozen vision transformers without the need for audio pre-training and (2) enabling seamless information exchange between the audio and visual modalities, all achieved with a limited set of additional trainable parameters. Through a lightweight bottleneck block on top of a simple cross-attention module that employs the $CLS$ token from both modalities as an intermediary for cross-modal information exchange, AV-PEA achieves robust audio-visual representations for several audio-visual tasks, including audio-visual event localization (AVEL), audio-visual question answering (AVQA), audio-visual retrieval (AVR), and audio-visual captioning (AVC). Encouragingly, comprehensive experimentation revealed that our AV-PEA achieves performance on par with or exceeding state-of-the-art methods. Furthermore, AV-PEA distinguishes itself with a consistent design and a uniform count of trainable parameters across diverse tasks, ensuring straightforward generalization for many audio-visual applications.

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

## 6  PyTorch code of the proposed AV-PEA

As explained in the paper, the proposed AV-PEA (audio-visual parameter-efficient adapter) is a streamlined adapter module crafted for fine-tuning pre-trained vision transformers in the context of audio-visual learning. Algorithm 1 provides the PyTorch code snippet for the AV-PEA, as mathematically presented in Section 3.3.

---

**Algorithm 1** PyTorch code for the AV-PEA adapter

---

```python
class AVPEA(nn.Module):
    """An audio-visual parameter-effecient adapter (AV-PEA)."""
    def __init__(self, in_dim, out_dim, reduction_factor=8):
        super().__init__()
        #scalar parameter
        self.gate = nn.Parameter(torch.tensor(0))# g in the equations 7 & 8
        self.q_gate = nn.Parameter(torch.tensor(0))# h in the equations 5 & 6

        #dimension reduction
        self.sampling_size = in_dim // reduction_factor

        #bottleneck block
        self.ln_before = nn.LayerNorm(in_dim)
        self.down_sampler = nn.Conv2d(in_dim, self.sampling_size, 1, bias=False)
        self.bn_before = nn.BatchNorm2d(self.sampling_size)
        self.activation = nn.ReLU(inplace=True)
        self.bn_after = nn.BatchNorm2d(out_dim)
        self.up_sampler = nn.Conv2d(self.sampling_size, out_dim, 1, bias=False)
        self.ln_after = nn.LayerNorm(out_dim)

    def forward(self,x, y=None):#e.g. x=audio input and y=visual input
        q = x[:,0:1,...]#the CLS token of modality x
        new_x = y
        k = new_x.permute(0,2,1)
        v = new_x

        #cross attention
        att = torch.bmm(q, k)
        att = F.softmax(att, dim=-1)
        res = torch.bmm(att, v)

        #
        res = x[:,0:1,...] + self.q_gate*res.contiguous()
        new_x = torch.cat((res,x[:,1:,...]), dim=1)

        #bottleneck block
        new_x = self.ln_before(new_x).permute(0,2,1).unsqueeze(-1) #normalization
        z = self.down_sampler(new_x)        #down_sampler
        z = self.bn_before(z)               #batch normalization
        z = self.activation(z)              #activation
        output = self.up_sampler(z)         #up_sampler
        output = self.bn_after(output)      #batch normalization
        output = self.ln_after(output.squeeze(-1).permute(0,2,1)) #normalization

        #
        output = self.gate * output

        return output
```

---

