# OpenReview forum: "AV-PEA: PARAMETER-EFFICIENT ADAPTER FOR AUDIO-VISUAL MULTIMODAL LEARNING"
_ICLR.cc/2024/Conference — ICLR 2024 Conference Withdrawn Submission_

### Official Review · Reviewer_diyf · 2023-10-25

**Soundness:** 2 fair
**Presentation:** 1 poor
**Contribution:** 2 fair
**Rating:** 3
**Confidence:** 3

**Summary:**

This paper describes a new lightweight adapter to extend existing pre-trained (on image-related tasks) ViT models for audio-visual tasks. In short, the authors feed audio and video into two frozen copies of the pre-trained ViT and, within each ViT block, add an AV-PEA (audio-visual parameter efficient adapter) that connects the two modalities. This is done efficiently by leveraging the CLS token from the source modality to learn from other other modality's tokens, rather than leveraging all tokens - this makes the process simpler and more flexible. The authors train their model for multiple tasks including audio-visual event localization,question-answering, retrieval, and captioning. They achieve state-of-the-art results on some tasks, and competitive performance on others, consistently outperforming their manning competitor - LAVisH.

**Strengths:**

Overall the idea of leveraging the CLS token to enable more efficient and adaptable fusion is a good one - the CLS token is a fundamental part of the ViT architecture and is designed to be a very powerful token within the latent space, so leveraging it explictly makes a lot of sense. Overall the idea is elegant and well-motivated. The motivation for few-parameter adapters rather than full-finetuning is not unique to this paper but remains strong.

The method clearly works - it outperforms LAVisH on all tasks from what I can understand. This is impressive given the simplicity of this change.

The paper is well-written in most parts and is clear regarding its results and their meaning.

**Weaknesses:**

Overall there are two major issues with this paper:

1. A lack of contribution. Fundamentally, the methodology presented here, down to the architecture, is extremely reminiscent of LAVisH, to the point where I really think the authors should explicitly consider themselves an extension of LAVisH. There is nothing wrong with this per say, but the differences here are only in the fact that the cross-attention between modalities is modified by explicitly leveraging the CLS token. This is a good change and makes sense but I don't think it's enough for a paper in the current format - I think many more ablations on the nature and impact of the CLS token, and strong explanations for its role using interesting visualizations, etc. would be necessary to justify this as its own paper. Also, a lot of the content seems inspired by LAVisH to an excessive extent - the same red and blue symbols are used in tables, which is somewhat acceptable I suppose, but Figure 1 for example is a bit too similar to the main figures in LAVisH in my opinion.

2. The presentation of the paper is poor. Figure 1 feels very rushed. It is aesthetically displeasing regarding the colors and shapes chosen (white blocks with a flame rather than incorporating it into the color of the box, among many other tasks), but also functionally problematic - the legend is strangely in the middle of the figure and is incomplete (e.,g., addition and multiplication symbols should be defined). The tables in this paper are also clearly lacking horizontal lines for clarity, and bold numbers to represent the best results for each column - this is extremely important to quickly understand their contents. Legends are also lacking, in particular for Table 4, where it does not describe what the accuracy means in that specific table, for example - tables should be self-contained and understandable by looking only at the caption.

Typos:
- CLS is used as an abbreviation at the end of Section 2, but only defined in Section 3.1.

**Questions:**

Will you be releasing 1. training code 2. inference code 3. pre-trained models publicly? This would be greatly appreciated as a contribution to the open-source community.

---

### Official Review · Reviewer_G8r2 · 2023-10-28

**Soundness:** 3 good
**Presentation:** 3 good
**Contribution:** 3 good
**Rating:** 6
**Confidence:** 4

**Summary:**

The authors propose a parameter efficient adapter method to fine-tune ViT and adapt to audio tasks, and show that it improves on audio-visual tasks such as audio-visual event localization, question answering, retrieval, and captioning.

**Strengths:**

- This work improves upon previous work LAVISH in both accuracy and parameter efficiency.
- The presentation is clear and the organization of the writing and narratives are easy to follow for the readers.

**Weaknesses:**

- One of the main claim in this line of work is to adapt Vision models pre-trained with ImageNet to audio, and all of the tasks selected here involve audio-visual. The ablation study shown in Table 4 include one entry with audio only stream, it would be great to see if this method can directly be applied to various audio tasks and how they compare with models trained with audio data only.
- Most of the results and discussions shown in 4.2 are to go through the same information listed in the table. It would provide more insights to the readers if there are some in depth error analysis and tradeoffs between the choice of LAVISH and AV-PEA.

**Questions:**

- In table 1, what are the difference between LAVISH encoders and CMBS? Would it make sense to provide another entry which only replace LAVISH with AV-PEA, similar to what is in table 2?
- In table 3, for both AVR and AVQA, why are adding efficient adapter techniques (both LAVISH and AV-PEA) hurting the performance compared to original VALOR? Dive deeper into the reasons here might be an interesting direction to further understand proposed methods.

---

### Official Review · Reviewer_Bn7A · 2023-10-30

**Soundness:** 3 good
**Presentation:** 2 fair
**Contribution:** 2 fair
**Rating:** 5
**Confidence:** 4

**Summary:**

This paper proposes a novel adapter structure, AV-PEA, to extend a Transformer-based visual encoder for 2D-spectrogram-image-based audio processing. The approach is tested based on several (non-speech) audio-visual benchmarks and achieves competitive results compared to existing approaches.

**Strengths:**

1. Audio-visual tasks are important, but currently understudied compared to image-text tasks.
2. The design of the adaptor structure makes sense and the results are promising.
3. The presentation of the paper is mostly clear.

**Weaknesses:**

1. It has been known for long that audio clips can be represented as 2D spectrogram images and achieve speech recognition and audio event detection using computer vision approaches, and many studies have done this. Therefore, the main novelty in this paper is to propose the AV-PEA adaptor, which may not be sufficient for an ICLR paper.

2. A fundamental weakness of extending an image encoder for audio input is that it can only handle audio sequences with a maximum length (sometimes even fixed length only). It also suffers from all weaknesses/assumptions that a short-term Fourier transform suffers (since it is used to generate the spectrograms)

3. The proposed AV-PEA was mostly compared to the existing LAVISH approach. AV-PEA requires a similar amount of parameters than LAVISH (12.9M vs 13.4M) and sometimes achieves slightly better performance (e.g. Table 1) but sometimes not (e.g. Table 2 and 3). The decreased 0.5M model parameters take only a tiny amount in the overall number of model parameters (27.2M + 325.6M)

4. In Tables 1 & 2, possibly better to use \citep rather than \citet, or at least add a "by" in between.

5. The authors sometimes followed a good fashion by quoting text-based functions with \text{} in equations (1)-(4), and sometimes didn't follow the same fashion in other equations, such as equations (5)-(8)

6. It seems to me the term "audio" used in this paper only covers sound events (non-speech or description of the speaker etc.), but not any speech content. This is very confusing to the readers and quite limited in applications. The authors should try to support speech or at least make this clear in the limitation section.

7. Some test data used in this paper, such as music AVQA, may need to be verified. There seem to have been overlapped video samples in its training and test sets.

**Questions:**

1. Some commonly used audio-visual (non-speech) benchmarks were not tested in this paper, which makes the results incomparable to other studies. Could the authors provide the results on at least these two datasets:
1).  Audio-visual scene-aware dialog: https://arxiv.org/abs/1901.09107
2). VGGSound: A large-scale audio-visual dataset: https://arxiv.org/abs/2004.14368

2. Although video tasks were mentioned many times in the paper, it is less clear to me how the proposed method can be applied to video tasks, which require temporal alignment between one audio embedding with many video-frame embeddings.

---

### Official Review · Reviewer_oLz4 · 2023-10-31

**Soundness:** 2 fair
**Presentation:** 2 fair
**Contribution:** 2 fair
**Rating:** 3
**Confidence:** 4

**Summary:**

This paper proposed an audio-visual parameter-efficient adapter (AV-PEA) designed to improve multimodal transfer learning for audio-visual tasks. This adaptor can be applied on a frozen vision transformer (ViT) to be adept at processing audio inputs without prior knowledge of audio pre-training. Because only the CLS token of each stream serves as an intermediary in the cross-attention module,  the computation cost is linear. The experimental result shows limited improvement compared to another adapter baseline (LAVISH).

**Strengths:**

1) The proposed method is simple and easy to follow with the PyTorch code snippet provided in the appendix.

2) Because only the CLS token of each stream serves as an intermediary in the cross-attention module, the quadratic computation  cost can be reduced to linear.

**Weaknesses:**

1)	The novelty is limited. The most interesting statement in this paper is about applying an adaptor on frozen ViT for different AV applications. However, it was proposed by a previous paper [1].  Hence the novelty of this paper is mainly based on using the CLS token in the cross-attention module.
2)	The performance improvement is limited compared to LAVISH [1].
3)	In [1], they can obtain higher scores using LAVISH with Swin-V2-L on  AVEL and AVQA tasks. Authors should also report the results of their AV-PEA with Swin-V2-L to compare with current SOTA.
4) Figure 1 and the equations (1)-(8) are unclear. I suggest putting the corresponding variables in the figure.

[1] Yan-Bo Lin, Yi-Lin Sung, Jie Lei, Mohit Bansal, and Gedas Bertasius. Vision transformers are
parameter-efficient audio-visual learners. In Proceedings of the IEEE/CVF Conference on Computer
Vision and Pattern Recognition, pp. 2299–2309, 2023.

**Questions:**

How does it perform when combining AV-PEA with Swin-V2-L?